# SmoothGNN: Smoothing-aware GNN for Unsupervised Node Anomaly Detection

## Abstract

The smoothing issue in graph learning leads to indistinguishable node representations, posing significant challenges for graph-related tasks. However, our experiments reveal that this problem can uncover underlying properties of node anomaly detection (NAD) that previous research has missed. We introduce Individual Smoothing Patterns (ISP) and Neighborhood Smoothing Patterns (NSP), which indicate that the representations of anomalous nodes are harder to smooth than those of normal ones. In addition, we explore the theoretical implications of these patterns, demonstrating the potential benefits of ISP and NSP for NAD tasks. Motivated by these findings, we propose SmoothGNN, a novel unsupervised NAD framework. First, we design a learning component to explicitly capture ISP for detecting node anomalies. Second, we design a spectral graph neural network to implicitly learn ISP to enhance detection. Third, we design an effective coefficient based on our findings that NSP can serve as coefficients for node representations, aiding in the identification of anomalous nodes. Furthermore, we devise a novel anomaly measure to calculate loss functions and anomalous scores for nodes, reflecting the properties of NAD using ISP and NSP. Extensive experiments on 9 real datasets show that SmoothGNN outperforms the best rival by an average of 14.66% in AUC and 7.28% in Average Precision, with 75x running time speedup, validating the effectiveness and efficiency of our framework.

## CCS Concepts

• **Computing methodologies → Anomaly detection**.

## Keywords

Unsupervised Node Anomaly Detection, Spectral GNN, Smoothing Patterns

**ACM Reference Format:**
Anonymous Author(s). 2024. SmoothGNN: Smoothing-aware GNN for Unsupervised Node Anomaly Detection. In *Woodstock '18: ACM Symposium on Neural Gaze Detection, June 03–05, 2018, Woodstock, NY*. ACM, New York, NY, USA, 12 pages. https://doi.org/XXXXXXX.XXXXXXX

## 1 Introduction

*Node anomaly detection (NAD)* aims to identify nodes in a graph that exhibit anomalous patterns compared to the majority of nodes

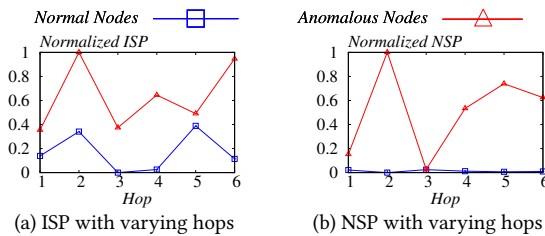

(a) ISP with varying hops     (b) NSP with varying hops

**Figure 1: Smoothing Patterns of Amazon.**

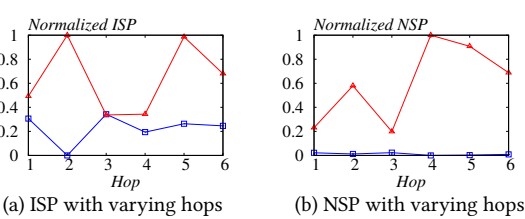

(a) ISP with varying hops     (b) NSP with varying hops

**Figure 2: Smoothing Patterns of T-Finance.**

[1, 21]. As the widespread prevalence of graph data driven by advances in modern technologies over the past three decades, NAD has become a trending topic due to its crucial role in various real-world applications, such as fraud detection in financial networks [13], malicious reviews detection in social networks [22], and hotspot detection in chip manufacturing [29].

However, the complicated information and large scale of real-world graphs present challenges in effectively and efficiently detecting anomalous nodes, especially in unsupervised settings [7, 11, 20]. To address these challenges, various designs have been proposed for the unsupervised NAD task, such as shallow models [18, 25], reconstruction models [15, 28], self-supervised models [8–10, 24], and special models [2, 12, 14, 26]. However, these methods usually face effectiveness or efficiency issues in real-world deployment for NAD tasks. To be specific, shallow models have limited expressiveness due to the hand-crafted rules, reconstruction models and self-supervised models are unlikely to be used in real applications due to high computational complexity, and special models face the challenge of finding an effective identifier of NAD.

To address these limitations, we re-evaluate the propagation procedure of NAD tasks and find that the smoothing issue can provide potential advantages for detecting anomalies in graphs. Specifically, we design two novel measures: *Individual Smoothing Patterns (ISP)* and *Neighborhood Smoothing Patterns (NSP)*, to analyze the smoothing issue from different perspectives. For ISP, we calculate the average normalized distances between node representations at each propagation hop and the converged representations obtained after an infinite number of hops for both anomalous and normal nodes. For NSP, we calculate the average normalized similarities within the neighborhoods of anomalous and normal nodes, respectively. Notably, these two smoothing patterns exhibit distinct behaviors across different types of nodes in real-world datasets like

 

Amazon and T-Finance, as illustrated in Figures 1 and 2, respectively. During propagation, the smoothing patterns of anomalous nodes generally exceed those of normal nodes at most hops. This observation provides a potential metric for assessing anomalous scores of nodes: the higher the smoothing patterns, the more likely a node is to be anomalous. Similar observations on other datasets can be found in Appendix A.3.

To explore the rationale behind this phenomenon, we conduct a theoretical analysis, revealing that the smoothing patterns are closely related to anomalous properties of nodes, originating from the structural and attribute information, as shown in Theorem 1. Besides, further exploration in Theorem 2 highlights the strong connection between the smoothing issue and the spectral space, providing insights for designing a spectral *Graph Neural Network (GNN)*. Moreover, as illustrated in Theorem 3, our findings indicate that NSP serves a role similar to spectral energy [6], which can be utilized as coefficients for node representations. Furthermore, we provide a theoretical guarantee in Theorem 4 to clarify the boundaries of the benefits derived from the smoothing issue.

Motivated by both experimental and theoretical findings, we introduce SmoothGNN, a novel graph learning framework for unsupervised NAD tasks. It consists of four key components: the *Smoothing-aware Learning Component (SLC)*, the *Smoothing-aware Spectral GNN (SSGNN)*, the *Smoothing-aware Coefficients (SC)*, and the *Smoothing-aware Measure (SMeasure)*. Specifically, SLC serves as a feature encoding module, explicitly capturing the ISP of anomalous and normal nodes, as supported by Theorem 1. Subsequently, based on Theorem 2, SSGNN is designed to learn node representations from the spectral space of the graph while implicitly capturing ISP to aid the learning process. Additionally, building upon Theorem 3, we design SC to extract both NSP and spectral energy information, providing complementary properties from other perspectives. Furthermore, through unifying the benefits of ISP and NSP proven by the empirical and theoretical results, we introduce a novel SMeasure to effectively and efficiently calculate the loss function and the anomalous scores. Ultimately, in contrast to previous studies in the unsupervised NAD area, such as [12, 26], which primarily focused on small or synthetic datasets, we conduct experiments on large-scale real datasets commonly encountered in practical applications, demonstrating the usefulness of SmoothGNN.

In summary, our work makes the following key contributions:

- To the best of our knowledge, we are the first to demonstrate the benefit of the smoothing issue on NAD tasks from both experimental and theoretical perspectives. Building upon this insight, we introduce a novel SMeasure as an anomaly measurement for unsupervised NAD tasks.
- We propose SmoothGNN, a novel framework that captures information from the smoothing process and spectral space of graphs, which can serve as a powerful backbone for NAD tasks.
- Our work stands out as the only one that conducts experiments on large real-world datasets for unsupervised NAD. Extensive experimental results showcase the effectiveness and efficiency of our proposed framework. Compared to state-of-the-art alternatives, SmoothGNN demonstrates superior performance in terms of AUC and Average Precision, with a speed-up of at least one order of magnitude.

## 2 Related Work

In recent years, unsupervised NAD has gained increasing interest within the graph learning community. Researchers have proposed a variety of models that can be broadly categorized into four groups: shallow models, reconstruction models, self-supervised models, and special models. Next, we briefly introduce several representative frameworks from these categories.

**Shallow Models**. Prior to the emergence of deep learning models, early works for the NAD tasks mainly focus on shallow models, which utilize statistical information and mathematical formulas to identify node anomalies. For instance, Radar [18] utilizes the residuals of attribute information and their coherence with graph information to identify anomalous nodes. ANOMALOUS [25] introduces a joint framework for NAD based on residual analysis. These models primarily adopt matrix decomposition and residual analysis, which inherently have limited capabilities in capturing the complex graph information, compared to deep learning models.

**Reconstruction Models**. Reconstruction models are prevalent approaches for unsupervised NAD, as the reconstruction errors of graph structures and node features inherently reflect the likelihood of a node being anomalous. Motivated by this, a prior work CLAD [15], proposes a label-aware reconstruction approach that utilizes Jensen-Shannon Divergence and Euclidean Distance. Besides, *graph auto-encoders (GAEs)* are widely adopted as reconstruction techniques. For example, GADNR [28] incorporates GAEs to reconstruct the neighborhood of nodes. Although previous studies have shown the usefulness of graph reconstruction, it is worth noting that reconstructing graph structures can be computationally expensive. Moreover, experimental findings [14] indicate that node feature reconstruction yields significant benefits for NAD. Therefore, a preferable choice is to focus exclusively on feature reconstruction to assist loss function, as introduced in the SmoothGNN framework.

**Self-supervised Models**. Aside from the reconstruction models, self-supervised models, such as contrastive learning frameworks, employ auxiliary tasks to guide unsupervised NAD. For example, NLGAD [10] constructs multi-scale contrastive learning networks to estimate the normality for nodes. Similarly, GRADATE [8] presents a multi-scale contrastive learning framework with subgraph-subgraph contrast to capture the local properties of nodes. Other examples include PREM [24] and ARISE [9], which employ node-subgraph contrast and node-node contrast to learn node representations, reflecting both local and global views of the graph. A recent study, TAM [26], leverages data augmentation to generate multi-view graphs, enabling the examination of the consistency of the local information within node neighborhoods. Based on the observation of local node affinity, TAM introduces a local affinity score to measure the probability of a node being anomalous, highlighting the importance of designing new measures for NAD. In contrast, SmoothGNN introduces the SMeasure to calculate anomalous scores, which utilizes a more flexible way to capture the anomalous properties of the nodes and requires fewer computational resources, enabling SmoothGNN to be applied to large-scale datasets.

**Special Models**. In addition to the above-mentioned models, there are also special models that leverage novel measurements to design loss or reward functions for calculating anomalous scores. For

instance, RAND [2] is the first work that leverages reinforcement learning in the unsupervised NAD task. It introduces an anomaly-aware aggregator to amplify messages from reliable neighbors. On the other hand, VGOD [14] presents a mixed-type framework that combines a reconstruction model and a self-supervised model. It incorporates a variance-based module to sample positive and negative pairs for contrastive learning, along with an attribute reconstruction module to reconstruct node features. Afterward, REC [12] utilizes a score-based generative model to boost the performance in this area. In contrast to these works, our SmoothGNN framework adopts a different strategy with theoretical analyses by utilizing feature reconstruction and the proposed SMeasure as the objective function, which achieves superior performance while requiring significantly less running time.

## 3 Preliminaries

**Notation**. Let $G = (A, X)$ denote a connected undirected graph with $n$ nodes and $m$ edges, where $X \in \mathbb{R}^{n \times F}$ represents node features and $A \in \mathbb{R}^{n \times n}$ represents the adjacency matrix. Let $A_{ij} = 1$ if there exists an edge between node $i$ and $j$, otherwise $A_{ij} = 0$. $D$ denotes the degree matrix. The adjacency matrix $\tilde{A}$ and degree matrix $\tilde{D}$ of graph $G$ with self-loops can be defined as $\tilde{A} = A + I_n$ and $\tilde{D} = D + I_n$, respectively, where $I_n \in \mathbb{R}^{n \times n}$ is an identity matrix. The Laplacian matrix $L$ is then defined as $L = I_n - \tilde{D}^{-\frac{1}{2}} \tilde{A} \tilde{D}^{-\frac{1}{2}}$. It can also be decomposed by $L = U \Lambda U^T$, where $U = (u_1, u_2, ..., u_n)$ represents orthonormal eigenvectors and the corresponding eigenvalues are sorted in ascending order, i.e. $\lambda_1 \leq ... \leq \lambda_n$. Let $x = (x_1, x_2, ..., x_n)^T \in \mathbb{R}^n$ be a signal on graph $G$, the graph convolution operation between a signal $x$ and a graph filter $g_\theta(\cdot)$ is then defined as $g_\theta(L) * x = U g_\theta(\Lambda) U^T x$, where the parameter $\theta \in \mathbb{R}^n$ is spectral filter coefficient vector. Table 6 in Appendix A.1 lists the frequently used notations in this paper.

**Unsupervised Node Anomaly Detection**. Let $V = \{v_1, ..., v_n\}$ denotes the node set of graph $G$, then unsupervised NAD tasks aim to learn an anomaly scoring function $f : V \to \mathbb{R}$, such that $f(v_n) < f(v_a)$, for $\forall v_n \in V_n$ and $\forall v_a \in V_a$, where $V_n$ and $V_a$ represents the normal and anomalous node set separately. In addition, due to the nature of the anomalies, it is typically assumed that $|V_n| \gg |V_a|$. Moreover, since this work focuses on the unsupervised setting, the class labels of the nodes during training are not accessible. Under such circumstances, unsupervised NAD tasks require effective and efficient techniques to help the learning of the framework.

**Spectral GNN**. Graph convolution operations [5, 16] can be approximated by the $T$-th order polynomial of Laplacians:

$$U g_\theta(\Lambda) U^T x \approx U \left( \sum_{t=0}^{T} \theta_t \Lambda^t \right) U^T x = \left( \sum_{t=0}^{T} \theta_t L^t \right) x,$$

where $\theta \in \mathbb{R}^{T+1}$ corresponds to polynomial coefficients. In the following Section 4.1, the prevalent graph convolution operation is demonstrated to have a strong relation with graph smoothing patterns. This key insight motivates the design of SmoothGNN, which can capture information from graph spectral space and anomalous properties behind smoothing patterns.

**Individual Smoothing Patterns**. As presented in a previous study [32], the node representations finally converge to a stable state, making it challenging to distinguish between different nodes. However, as discussed in Section 1, the distances between node representations at each propagation hop and the converged representations obtained after an infinite number of hops exhibit different patterns for anomalous and normal nodes. Hence, ISP can be denoted as:

$$I(x) = \left\| (P^t - P^\infty) x \right\|_2^2,$$

where $P^t$ is the propagation matrix after $t$ hops of propagation, $P^\infty$ is the converged state, and $x$ is the graph signal. ISP effectively describes the smoothing patterns of each individual node during propagation, as indicated by its definition. Subsequent analyses in Section 4.1 illustrate the effectiveness of ISP in NAD tasks, which can capture both spectral information and smoothing patterns.

**Neighborhood Smoothing Patterns**. To describe the smoothing patterns from a different perspective, we adopt the concept of Dirichlet Energy [33] to define NSP as follows:

$$N(x^t) = \sum_{i,j=1}^{n} a_{i,j} \left\| \frac{x_i^t}{\sqrt{d_i + 1}} - \frac{x_j^t}{\sqrt{d_j + 1}} \right\|_2^2,$$

where $a_{i,j}$ represents the $(i, j)$-th entry of the adjacency matrix $\tilde{A}$, $d_i$ is the degree of node $i$, and $x^t = P^t x$. According to this definition, NSP measures the similarities between neighboring nodes, indicating the smoothing patterns within neighborhoods during propagation. To explore the benefits of NSP, we delve into it in Section 4.1, revealing that NSP exhibits a strong correlation with spectral space and can serve as coefficients for node representations.

A detailed theoretical analysis of ISP and NSP can be found in Section 4.1, which supports the empirical results in Section 1 and motivates our design of SmoothGNN.

## 4 Method: SmoothGNN

Our observation in Section 1 highlights the different smoothing patterns exhibited by anomalous and normal nodes. In the following sections, we present detailed theoretical analyses and the design of our SmoothGNN. Specifically, Section 4.1 provides a comprehensive theoretical analysis of smoothing patterns, which serves as the motivation behind the design of two key components: SLC and SSGNN, to be elaborated in Sections 4.2 and 4.3, respectively. Moreover, our theoretical analysis in Section 4.1 reveals that the spectral energy of the graph can be represented by NSP, which inspires us to employ it as effective coefficients for node representations, to be detailed in Section 4.4. Finally, Section 4.5 illustrates the overall objective function, including a feature reconstruction loss and the proposed SMeasure.

## 4.1 Theoretical Analysis of Smoothing Patterns

The smoothing issue has been extensively studied in graph learning. However, previous studies such as [32] primarily focus on its negative aspects. This motivates us to explore the potential positive implications of the smoothing issue. To this end, we conduct a detailed analysis to demonstrate how ISP and NSP can reveal anomalous properties of nodes. All proofs of our theorems can be found in Appendix A.2.

Based on previous research [11, 26], both local views, such as neighboring nodes and their features, and global views, which encompass statistical information of entire graphs, contribute to the detection of node anomalies. The following Theorem 1 indicates that ISP can be represented by an augmented propagation matrix that incorporates both local and global information, suggesting that ISP can be an effective identifier for NAD tasks.

THEOREM 1. *Let $P = \frac{I_n + \tilde{A}}{2}$ denote the propagation matrix given the adjacency matrix $\tilde{A}$. For an augmented propagation matrix $B^t = (P - P^\infty)^t$, where $P^\infty$ represents the converged status of $P$, we can derive $B^t = P^t - P^\infty$ with $(i, j)$-th entry*

$$B_{i,j} = \frac{(2m+n)(\mathbb{I}[i=j]\sqrt{d_i+1} + 2a_{ij}) - 2(d_i+1)\sqrt{d_j+1}}{2\sqrt{d_i+1}(2m+n)},$$

*where $\mathbb{I}[\cdot]$ is the indicator function, $a_{i,j}$ is the $(i, j)$-th entry of the adjacency matrix, $d_i$ is the degree of node $i$, and $m, n$ represent the number of edges and nodes, respectively.*

Theorem 1 shows that when the graph signal $x$ propagates on the augmented propagation matrix $B$, the resulting node representation becomes aware of individual node features and local information, such as edge connections and the degrees of neighbors. Moreover, this matrix not only propagates graph signal through edges but also assigns the signal a transformation of statistical information of graph as coefficients, functioning similarly to an attention mechanism. It highlights the disparities arising from global views. Consequently, the augmented propagation matrix provides a more precise indication of the underlying properties of both anomalous and normal nodes compared to the original matrix. This observation is further supported by the empirical evidence of ISP shown in Section 1. Specifically, the comprehensive information contained in the augmented propagation matrix helps to elucidate the different smoothing processes of anomalous and normal nodes, where anomalous nodes are harder to converge than normal ones. Therefore, we employ this matrix in the design of the Smoothing-aware Learning Component (SLC) in Section 4.2.

In addition to the close relationship between the augmented propagation matrix and graph anomalies established by Theorem 1, previous studies [6, 31] have also shown a strong connection between graph anomalies and the graph spectral space. This motivates us to further investigate the relationships between the augmented propagation matrix and the graph spectral space. The following theorem confirms that column vectors of the augmented propagation matrix can be represented by a polynomial combination of graph convolution operations, indicating a strong correlation between the augmented propagation matrix and the graph spectral space.

THEOREM 2. *The augmented propagation matrix $B$ after $t$ hops of propagation can be expressed by $b^t = \sum_{k=0}^{t} \tilde{\theta}_k L^k uv$, where $b^t$ is a column vector of $B^t$, $\tilde{\theta}_k \in \mathbb{R}^n$ is the spectral filter coefficients, and $u, v$ represent the linear combinations of the eigenvectors of $\tilde{A}$ and $L$, respectively.*

Theorem 2 illustrates the connection between the augmented propagation matrix and the graph spectral space. This insight motivates us to design a Smoothing-aware Spectral Graph Neural Network (SSGNN) that not only leverages spectral information

but also captures ISP, to be elaborated in Section 4.3. Besides, previous work [31] has shown spectral energy (refer to Definition 1) can serve as an effective identifier for NAD tasks. Given our findings that reveal a strong connection between smoothing patterns and spectral space, we further investigate the relationship between smoothing patterns and spectral energy. First, we provide the definition of spectral energy:

DEFINITION 1 ([6, 31]). *Given the graph Laplacian matrix $L = U\Lambda U^T$ and a graph signal $x$, the graph Fouier Transformation of $x$ is defined as $\hat{x} = \{\hat{x_1}, \cdots, \hat{x_n}\} = U^T x$. The spectral energy of the graph at $\lambda_k$ can be expressed as $\frac{\hat{x}_k^2}{\sum_{i=1}^{n} \hat{x}_i^2}$.*

Based on Definition 1, we present the following theorem, which shows that NSP can serve a similar role as spectral energy.

THEOREM 3. *Given a graph $G$ with Laplacian matrix $L$ and a graph signal $x$, NSP can be represented by $N(x) = \frac{x^T L x}{x^T x} = \frac{\sum_{j=1}^{n} \lambda_j \hat{x}_j^2}{\sum_{i=1}^{n} \hat{x}_i^2}$, where the $\lambda_j$ is the $j$-th eigenvalue of $L$.*

Theorem 3 shows that the NSP of nodes can be represented by a polynomial combination of the spectral energy, indicating that NSP can serve as an effective identifier for NAD tasks. Recap from Section 3 that NSP characterizes the smoothing patterns within neighborhoods of nodes, which is complementary to the previous local view depicted by ISP. This motivates us to combine ISP and NSP to derive final representations and establish a novel measure for the anomaly scoring function. Specifically, we introduce Smoothing-aware Coefficients (SC) as the coefficients of node representations to facilitate the identification of different nodes, and Smoothing-aware Measure (SMeasure) as the metric for calculating the anomalous scores, which will be discussed in Sections 4.4 and 4.5, respectively.

So far, we have introduced the intuition behind four key components of SmoothGNN from both empirical and theoretical perspectives. Beyond the design, we further analyze the maximum propagation hops of SmoothGNN that do not provide additional information for NAD tasks. To be specific, if the current node representations have reached a converged state, additional layers of SmoothGNN will not yield substantial benefits but will consume extra computational resources. Therefore, we determine the layers of SmoothGNN based on Theorem 4. To achieve this, we provide $\epsilon$-smoothing [27] and illustrate the theorem of the converged hop.

DEFINITION 2 ([27]). *For any GNN, we call it suffers from $\epsilon$-smoothing if and only if after $T$ hops of propagation, the resulting feature matrix $H^t$ at hop $t \geq T$ has a distance no larger than $\epsilon$ with respect to a subspace $S$, namely, $d_S(H^t) \leq \epsilon, \forall t \geq T$, where $d_S(H^t) := min_{M \in S} ||H^t - M||_F$ represents the Frobenius norm from $H^t$ to the subspace $S$.*

THEOREM 4. *Given the subspace $S$ with threshold $\epsilon$, a GNN model will suffer from $\epsilon$-smoothing issue when the propagation hop $t = \left\lceil \frac{\log(\epsilon/d_S(X))}{\log(\tau\lambda)} \right\rceil$, where $\tau$ is the largest singular value of the graph filters over all layers, $\lambda$ is the second largest eigenvalue of the propagation matrix, and $X$ is the feature matrix of graph $G$.*

Theorem 4 provides a theoretical guarantee regarding the maximum propagation hops that can contribute to the learning process, which provides guidance for choosing the appropriate number of

layers in our experiments, as shown in Section 5 and Appendix 5.4. In the following Sections 4.2, 4.3, 4.4, and 4.5, we elaborate on our SmoothGNN framework in detail.

## 4.2 Smoothing-aware Learning Component

Motivated by Theorem 1, we propose a simple yet powerful component to explicitly capture the ISP of nodes. Specifically, we first calculate the augmented propagation matrix $B^t = P^t - P^\infty$ for $t = 0, \cdots, T$. Next, we employ a set of $(T + 1)$ MLPs to obtain the latent node representations propagated on each $B^t$. Finally, an additional MLP is adopted to fuse the node representations obtained from $(T + 1)$ propagation hops. Let $\tilde{X}_t$ denote the node features $X$ after the $t$-th feature transformation, the representation of the $i$-th node in SLC can be expressed as:

$$h_i^{SLC} = \text{MLP}(\text{CONCAT}((B^0 \tilde{X}_0)_i, \cdots, (B^T \tilde{X}_T)_i)).$$

Despite the simplicity of the SLC module, it can capture the information underlying the ISP of different nodes and thus can serve as an effective component for unsupervised NAD tasks as shown in the later experiments.

In addition to explicitly learning from ISP, capturing information from the graph topology and node features can also be useful to NAD. The combination of explicit and implicit learning enables the collection of comprehensive information required for NAD tasks, which is demonstrated in the ablation study in Section 5.3. The details of implicit learning GNN are presented as follows.

## 4.3 Smoothing-aware Spectral GNN

As stated in Theorem 2, the column vector of augmented propagation matrix after $t$ hops of propagation can be represented as $b^t = \sum_{t=0}^T \tilde{\theta}_t L^t uv$, demonstrating the capability of the graph spectral space to reveal underlying node properties for NAD. This motivates our design of a spectral GNN to learn node representations. Based on the theoretical analysis, employing a polynomial combination of graph spectral filters as the graph convolution operation can be a natural choice. To maintain the simplicity of our framework, we leverage $T$-th order polynomial of graph Laplacian as the backbone filter. Specifically, let $g(X)_T$ be the graph convolution operation, we have:

$$g(X)_T = (\sum_{t=0}^T \theta_t L^t) X.$$

Similar to SLC, we consider $\tilde{X}_t$ as node features after $t$-th feature transformation for each graph convolution operation. Subsequently, we employ an MLP to fuse the spectral node representations obtained from each propagation hop to generate final node representations. The representation of $i$-th node can be expressed as:

$$h_i^{GNN} = \text{MLP}(\text{CONCAT}((g(\tilde{X}_0)_0)_i, \cdots, (g(\tilde{X}_T)_T)_i)).$$

Note that we utilize shared weights in SLC and SSGNN, so that the learnable weights can be influenced by both components simultaneously, which makes the assistance of feature reconstruction for SMeasure in Section 4.5 more effective. By incorporating these two components, our framework can capture information from both spectral space and smoothing patterns.

In addition, as discussed in Section 4.1, combining ISP and NSP will enable the framework to effectively distinguish anomalous nodes and normal nodes. Hence, we utilize NSP as the coefficients for SLC and SSGNN components to achieve this goal. The details of SC will be further introduced in Section 4.4.

## 4.4 Smoothing-aware Coefficients

Theorem 3 shows that NSP can be interpreted as a polynomial combination of spectral energy, which is an effective identifier of NAD tasks as shown in previous works [6, 31]. Motivated by the results, we design SC as coefficients for node representations. Specifically, we calculate the linear combination of $(T + 1)$ hops of NSP based on Theorem 3, which can be expressed as:

$$SC(X) = diag\left(\frac{X^T L X}{X^T X}\right),$$

$$\alpha = \sigma(\text{MLP}(\text{CONCAT}(SC(P^0 \tilde{X}_0), \cdots, SC(P^T \tilde{X}_T)))),$$

where $diag(\cdot)$ denotes the diagonal entries of a square matrix, and $\sigma(\cdot)$ is the Sigmoid function. Then, we utilize element-wise multiplication $*$ to modify representations $h_i^{SLC}$ and $h_i^{GNN}$:

$$h_i^{SCSLC} = h_i^{SLC} * \alpha, h_i^{SCGNN} = h_i^{GNN} * \alpha.$$

The final representations generated by SLC and SSGNN with the assistance of SC are utilized to calculate the loss function and SMeasure, which will be illustrated in the following section.

## 4.5 Smoothing-aware Measure

According to previous work [14], feature reconstruction loss can assist in learning effective measures for NAD. This inspires us to design a loss function combined with two components: the feature reconstruction loss and SMeasure. For the feature reconstruction loss, we use $h_i^{SCGNN}$ to reconstruct the original feature:

$$L_{con} = \frac{1}{n} \sum_{i=1}^n ||h_i^{SCGNN} - x_i||_2,$$

where $x_i$ is the $i$-th row of the original feature matrix $X$. For SMeasure, we leverage the representations obtained from SLC as it naturally captures the underlying properties in the smoothing patterns. To be specific, we define SMeasure as follows:

$$f_{smooth}(h_i^{SCSLC}) = \sigma(\text{AVG}(h_i^{SCSLC})),$$

where $\sigma(\cdot)$ represents the Sigmoid function, and $\text{AVG}(\cdot)$ represents the column-wise average function. Based on SMeasure, we further define the smoothing-aware loss function:

$$L_{smooth} = \frac{1}{n} \sum_{i=1}^n f_{smooth}(h_i^{SCSLC}).$$

The final loss function is a combination of both $L_{con}$ and $L_{smooth}$:

$$L = L_{con} + L_{smooth}.$$

The loss function is carefully designed to leverage the feature reconstruction loss to facilitate the learning process of SMeasure. This combination enables the loss function to capture valuable information from the smoothing patterns and the reconstruction errors across different nodes, which can be demonstrated in Section 5.3.

**Table 1: Statics of 9 real-world datasets, including the number of nodes and edges, the node feature dimension, the average degree of nodes, and the ratio of anomalous labels.**

| Categories | Datasets | #Nodes | #Edges | #Feature | Avg. Degree | Anomaly Ratio |
|---|---|---|---|---|---|---|
| Small | Reddit | 10,984 | 168,016 | 64 | 15.30 | 3.33% |
| | Tolokers | 11,758 | 519,000 | 10 | 44.14 | 21.82% |
| | Amazon | 11,944 | 4,398,392 | 25 | 368.25 | 6.87% |
| Medium | T-Finance | 39,357 | 21,222,543 | 10 | 539.23 | 4.58% |
| | YelpChi | 45,954 | 3,846,979 | 32 | 83.71 | 14.53% |
| | Questions | 48,921 | 153,540 | 301 | 3.14 | 2.98% |
| Large | Elliptic | 203,769 | 234,355 | 167 | 1.15 | 9.76% |
| | DGraph-Fin | 3,700,550 | 4,300,999 | 17 | 1.16 | 1.27% |
| | T-Social | 5,781,065 | 73,105,508 | 10 | 12.65 | 3.01% |

Minimizing this loss function empowers the model to effectively reduce the ratio of anomalous nodes in the predicted results, thereby addressing the challenge of extremely unbalanced data in NAD. With this comprehensive loss function, our SmoothGNN can optimize the shared weights of two key components. Consequently, our framework excels in learning more accurate node representations for the task of detecting anomalous nodes.

## 5 Experiments

### 5.1 Experimental Setup

**Datasets.** We evaluate SmoothGNN on 9 real-world datasets, including Reddit, Tolokers, Amazon, T-Finance, YelpChi, Questions, Elliptic, DGraph-Fin, and T-Social. These datasets are obtained from the benchmark paper [30], consisting of various types of networks and corresponding anomalous nodes. Based on their number of nodes, we divide these datasets into three categories, Small, Medium, and Large, as shown in Table 1. Note that, unlike previous works in the unsupervised NAD area, we only utilize real-world datasets with a sufficient number of nodes. To the best of our knowledge, SmoothGNN is the only model in this field that conducts comprehensive experiments on large-scale datasets such as T-Social to validate the efficiency and effectiveness of various models.

**Baselines.** We compare SmoothGNN against 11 state-of-the-art competitors, including shallow models, reconstruction models, self-supervised models, and special models.
- Shallow models: RADAR [18], and ANOMALOUS [25].
- Reconstruction models: CLAD [15] and GADNR [28].
- Self-supervised models: NLGAD [10], GRADATE [8], PREM [24], ARISE [9], and TAM [26].
- Special models: RAND [2], VGOD [14], and REC [12].

**Experimental Settings**. In line with the experimental settings of prior studies, such as [14, 19, 26], we conduct transductive experiments on these datasets. The parameters of SmoothGNN are set according to the categories of the datasets. The specific parameters for each category can be found in Appendix A.5. To ensure a fair comparison, we obtain the source code of all competitors from GitHub and execute these models using the default parameter settings suggested by their authors.

**Comparison Metrics**. To provide fair comparison results, we follow previous works in this area, utilizing AUC and Average Precision (AP) as the metrics for comparison. Specifically, AUC provides an aggregate measure of performance across all possible classification thresholds. One way of interpreting AUC is the probability that the model ranks a random positive example more highly than a random negative example. AP provides insights into the precision of anomaly detection at all decision thresholds. It calculates the area under the Precision-Recall curve, which balances the effects of precision and recall. A higher AP indicates a lower false-positive rate and false-negative rate. As a result, if a framework can achieve higher AUC and AP than other frameworks, it is comprehensive enough to show that such a framework is effective for unsupervised NAD tasks. Moreover, we also report the running time cost to demonstrate the efficiency of our framework.

### 5.2 Main Results

We evaluate the performance of SmoothGNN against different state-of-the-art competitors in the field of unsupervised NAD. Table 2 reports the AUC and AP scores of each model across 9 datasets. The best result on each dataset is highlighted in boldface. Our key observations are as follows.

Firstly, most existing unsupervised NAD models struggle to handle large datasets, with only VGOD, REC, and the proposed SmoothGNN successfully running on the two largest datasets. This highlights the need for the development of unsupervised models capable of handling large-scale datasets.

Shallow models, Radar and ANOMALOUS, apply residual analysis to solve NAD, which poses challenges in capturing the underlying anomalous properties from a spectral perspective. In comparison, SmoothGNN takes the lead by 29.37% and 29.03% in terms of AUC, and 11.52% and 11.63% in terms of AP on average across 6 datasets, respectively. Moreover, these shallow models are unable to handle the 3 large datasets due to memory constraints. These results demonstrate that shallow models are both time-consuming and ineffective when applied to real-world NAD datasets.

Next, we examine reconstruction models, CLAD and GADNR, which utilize reconstruction techniques to detect graph anomalies. While these models leverage both structure and feature reconstruction to calculate the anomalous score for each node, they fail to utilize a more effective identifier, such as smoothing patterns, leading to inferior performance. SmoothGNN outperforms these models by 26.14% and 17.75% in terms of AUC, and 10.31% and 8.30% in terms of AP on average across different datasets, respectively.

We then compare SmoothGNN with self-supervised models, NLGAD, GRADATE, PREM, ARISE, and TAM. Although self-supervised

**Table 2: AUC and Precision (%) on 9 datasets, where "-" represents failed experiments due to memory constraint. The best result on each dataset is highlighted in boldface.**

| Datasets | Metrics | Shallow | | Reconstruction | | Self-supervised | | | | | Special | | | |
|---|---|---|---|---|---|---|---|---|---|---|---|---|---|---|
| | | RADAR | ANOMALOUS | CLAD | GADNR | NLGAD | GRADATE | PREM | ARISE | TAM | RAND | VGOD | REC | SmoothGNN |
| Reddit | AUC | 0.4372 | 0.4481 | 0.5784 | 0.5532 | 0.5380 | 0.5261 | 0.5518 | 0.5273 | 0.5729 | 0.5417 | 0.4931 | 0.5510 | **0.5946** |
| | AP | 0.0273 | 0.0309 | **0.0502** | 0.0373 | 0.0415 | 0.0393 | 0.0413 | 0.0402 | 0.0425 | 0.0356 | 0.0324 | 0.0421 | 0.0438 |
| Tolokers | AUC | 0.3625 | 0.3706 | 0.4061 | 0.5768 | 0.4825 | 0.5373 | 0.5654 | 0.5514 | 0.4699 | 0.4377 | 0.4988 | 0.4314 | **0.6870** |
| | AP | 0.1713 | 0.1731 | 0.1921 | 0.2991 | 0.2025 | 0.2364 | 0.2590 | 0.2505 | 0.1963 | 0.1939 | 0.2212 | 0.1946 | **0.3517** |
| Amazon | AUC | 0.2318 | 0.2318 | 0.2026 | 0.2608 | 0.5425 | 0.4781 | 0.2782 | 0.4782 | 0.8028 | 0.3585 | 0.5182 | 0.5869 | **0.8408** |
| | AP | 0.0439 | 0.0439 | 0.0401 | 0.0424 | 0.0991 | 0.0634 | 0.0744 | 0.0677 | 0.3322 | 0.0492 | 0.0779 | 0.1349 | **0.3953** |
| T-Finance | AUC | 0.2824 | 0.2824 | 0.1385 | 0.5798 | 0.5231 | 0.4063 | 0.4484 | 0.4667 | 0.6901 | 0.4380 | 0.4814 | 0.5239 | **0.7556** |
| | AP | 0.0295 | 0.0295 | 0.0247 | 0.0542 | 0.0726 | 0.0376 | 0.0391 | 0.0393 | 0.1284 | 0.0403 | 0.0454 | 0.0454 | **0.1408** |
| YelpChi | AUC | 0.5261 | 0.5272 | 0.4755 | 0.4704 | 0.4981 | 0.4920 | 0.4900 | 0.4834 | 0.5487 | 0.5052 | 0.4878 | 0.5134 | **0.5758** |
| | AP | 0.1822 | 0.1700 | 0.1284 | 0.1395 | 0.1469 | 0.1447 | 0.1378 | 0.1415 | 0.1733 | 0.1470 | 0.1345 | 0.1623 | **0.1823** |
| Questions | AUC | 0.4963 | 0.4965 | 0.6207 | 0.5875 | 0.5428 | 0.5539 | 0.6033 | 0.6241 | 0.5042 | 0.6164 | 0.5075 | 0.4988 | **0.6444** |
| | AP | 0.0279 | 0.0279 | 0.0512 | 0.0577 | 0.0348 | 0.0350 | 0.0430 | **0.0619** | 0.0395 | 0.0442 | 0.0299 | 0.0279 | 0.0592 |
| Elliptic | AUC | - | - | 0.4192 | 0.4001 | 0.4977 | - | 0.4978 | - | - | - | 0.5723 | **0.5848** | 0.5729 |
| | AP | - | - | 0.0807 | 0.0778 | 0.1009 | - | 0.0905 | - | - | - | 0.1256 | **0.1337** | 0.1161 |
| DGraph-Fin | AUC | - | - | - | - | - | - | - | - | - | - | 0.5456 | 0.4710 | **0.6499** |
| | AP | - | - | - | - | - | - | - | - | - | - | 0.0148 | 0.0112 | **0.0199** |
| T-Social | AUC | - | - | - | - | - | - | - | - | - | - | 0.5999 | 0.0793 | **0.7034** |
| | AP | - | - | - | - | - | - | - | - | - | - | 0.0351 | 0.0157 | **0.0631** |

**Table 3: Running time (s) on 9 datasets, where "-" represents failed experiments due to memory constraint. The best result on each dataset is highlighted in boldface.**

| Datasets | Shallow | | Reconstruction | | Self-supervised | | | | | Special | | | |
|---|---|---|---|---|---|---|---|---|---|---|---|---|---|
| | RADAR | ANOMALOUS | CLAD | GADNR | NLGAD | GRADATE | PREM | ARISE | TAM | RAND | VGOD | REC | SmoothGNN |
| Reddit | 55.57 | 42.25 | 11.14 | 692.66 | 10886.19 | 7562.59 | 73.52 | 1261.99 | 5050.89 | 310.11 | 39.86 | 83.23 | **7.02** |
| Tolokers | 57.51 | 40.94 | 52.91 | 861.95 | 10504.91 | 7824.63 | 74.80 | 1281.71 | 5668.91 | 367.03 | 177.42 | 161.54 | **6.99** |
| Amazon | 42.79 | 38.07 | 431.20 | 2048.72 | 10649.83 | 7856.41 | 130.57 | 1267.38 | 1148.24 | 593.75 | 1517.70 | 2558.55 | **7.19** |
| T-Finance | 500.19 | 360.97 | 2161.16 | 14255.00 | 35648.72 | 30341.65 | 266.33 | 4223.50 | 81238.60 | 6746.10 | 5998.08 | 83339.64 | **16.69** |
| YelpChi | 730.81 | 513.83 | 418.95 | 5046.51 | 42435.07 | 35938.21 | 308.68 | 5042.99 | 102232.07 | 6588.60 | 1283.10 | 978.22 | **19.37** |
| Questions | 1205.05 | 1114.22 | 52.65 | 2795.99 | 51270.03 | 44235.87 | 409.45 | 6135.88 | 11603.81 | 7364.07 | 86.12 | 482.32 | **32.68** |
| Elliptic | - | - | 421.17 | 12568.50 | 193304.73 | - | 2149.77 | - | - | - | 231.63 | 566.79 | **205.10** |
| DGraph-Fin | - | - | - | - | - | - | - | - | - | - | 3420.84 | 6795.90 | **2924.99** |
| T-Social | - | - | - | - | - | - | - | - | - | - | 22984.10 | 80388.98 | **4877.05** |

models can boost the performance of unsupervised frameworks, their high memory requirements and computational costs make them prohibitive for large datasets. While NLGAD and PREM utilize sparse techniques to address these issues, they still cannot run on the two largest datasets. In comparison, SmoothGNN achieves an improvement of 14.95% and 17.66% in terms of AUC, and 8.44% and 8.63% in terms of AP on average across 7 datasets, respectively. Besides, SmoothGNN also outperforms GRADATE and ARISE by 18.41% and 16.12% in terms of AUC, and 10.28% and 9.53% in terms of AP on average across 7 datasets, respectively. In addition, TAM is the best rival in terms of performance, but its high memory usage and running time make it unable to run on large datasets. Across 6 datasets, SmoothGNN surpasses TAM by 8.49% in terms of AUC and 4.35% in terms of AP on average.   Finally, we examine the results of special models, RAND, VGOD and REC. RAND represents a novel direction for unsupervised NAD tasks but fails to leverage more advanced properties, such as smoothing patterns, to guide the learning process. As a result, SmoothGNN outperforms RAND by 20.01% in terms of AUC, and 11.05% in terms of AP on average

across 6 datasets. On the other hand, VGOD and REC are the only two competitors capable of running on all the datasets, demonstrating the benefits of designing efficient measures for unsupervised NAD tasks. However, SmoothGNN leverages similar techniques with a novel measure more efficiently and effectively, surpassing VGOD and REC by 14.66% and 19.82% in terms of AUC, and 7.28% and 6.72% in terms of AP on average across all datasets, respectively. Moreover, with VGOD as the most efficient and effective competitor, our SmoothGNN outperforms it in all datasets with a 75x speed-up in running time, which demonstrates the usefulness of our framework.

## 5.3 Ablation Study

The ablation study for SC is presented in Table 4. Notably, without SC to rearrange the weights of different dimensions in the spectral space, the performance drops significantly compared to the original SmoothGNN, which demonstrates the utilization of SC can boost the performances. It also underscores that capturing the smoothing patterns from different views will help the learning of the node

Table 4: Ablation study.

| Datasets | Reddit | | Tolokers | | Amazon | | T-Finance | | YelpChi | | Questions | | Elliptic | | DGraph-Fin | | T-Social | |
|---|---|---|---|---|---|---|---|---|---|---|---|---|---|---|---|---|---|---|
| Metrics | AUC | AP | AUC | AP | AUC | AP | AUC | AP | AUC | AP | AUC | AP | AUC | AP | AUC | AP | AUC | AP |
| SmoothGNN | 0.5946 | 0.0438 | 0.6870 | 0.3517 | 0.8408 | 0.3953 | 0.7556 | 0.1408 | 0.5758 | 0.1823 | 0.6444 | 0.0592 | 0.5729 | 0.1161 | 0.6499 | 0.0199 | 0.7034 | 0.0631 |
| w/o SC | 0.5437 | 0.0356 | 0.6115 | 0.2967 | 0.5131 | 0.0645 | 0.2869 | 0.0292 | 0.5715 | 0.1770 | 0.6260 | 0.0630 | 0.5596 | 0.1076 | 0.5868 | 0.0161 | 0.6639 | 0.0622 |
| w/o $L_{con}$ | 0.5801 | 0.0494 | 0.6645 | 0.3168 | 0.8106 | 0.3031 | 0.7311 | 0.0858 | 0.5608 | 0.1719 | 0.6335 | 0.0506 | 0.5655 | 0.1145 | 0.6189 | 0.0181 | 0.6715 | 0.0514 |

Table 5: AUC and Precision (%) on 8 datasets of SmoothGNN and SmoothGNN-A. Due to the high computational cost of SmoothGNN-A, we omit the results on the largest T-Social dataset.

| Datasets | Reddit | | Tolokers | | Amazon | | T-Finance | | YelpChi | | Questions | | Elliptic | | DGraph-Fin | |
|---|---|---|---|---|---|---|---|---|---|---|---|---|---|---|---|---|
| Metrics | AUC | AP | AUC | AP | AUC | AP | AUC | AP | AUC | AP | AUC | AP | AUC | AP | AUC | AP |
| SmoothGNN | 0.5946 | 0.0438 | 0.6870 | 0.3517 | 0.8408 | 0.3953 | 0.7556 | 0.1408 | 0.5758 | 0.1823 | 0.6444 | 0.0592 | 0.5729 | 0.1161 | 0.6499 | 0.0199 |
| SmoothGNN-A | 0.5919 | 0.0486 | 0.6731 | 0.3340 | 0.8008 | 0.2719 | 0.7408 | 0.1099 | 0.5697 | 0.1887 | 0.6388 | 0.0527 | 0.5695 | 0.1136 | 0.5893 | 0.0164 |

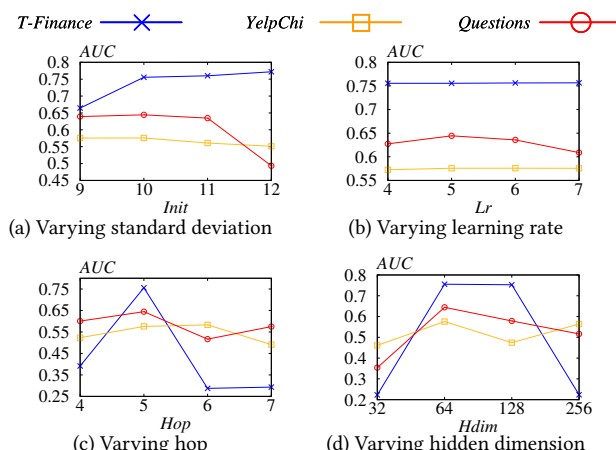

(a) Varying standard deviation

(b) Varying learning rate

(c) Varying hop

(d) Varying hidden dimension

Figure 3: Varying the standard deviation, learning rate, hop, and hidden dimension.

representations for NAD tasks. Moreover, without the assistance of feature reconstruction in the loss function, the performance of SmoothGNN will also drop to some extent as shown in Table 4, which proves the benefits of feature reconstruction as the assistance for the learning process. This phenomenon matches the results in previous works such as [14], highlighting the rationality of utilizing feature reconstruction in our framework.

## 5.4 Parameter Analysis

Next, we conduct experiments to analyze the effect of representative parameters: the standard deviation of weight initialization, the learning rate, the number of propagation hops, and the hidden dimension of SmoothGNN on T-Finance, YelpChi, and Questions datasets. Figure 3 reports the AUC of SmoothGNN as we vary the standard deviation from 9e-3 to 12e-3, the learning rate from 4e-4 to 7e-4, the hop from 4 to 7, and the hidden dimension from 32 to 256. As we can observe, when we set the standard deviation to 10e-3, SmoothGNN achieves relatively satisfactory performances across these three datasets. In terms of learning rate, SmoothGNN exhibits relatively stable performance, but we can identify an optimal one, so we set the learning rate to 5e-5. Meanwhile, SmoothGNN shows

a relatively stable and high performance in terms of all three presented datasets when we set the hop to 5. As a result, the hop is set to 5 in SmoothGNN. Besides, when setting the hidden dimension to 64, our SmoothGNN achieves the best performance. Hence, the hidden dimension in experiments are set to 64.

## 5.5 Alternative Smoothing Patterns

In addition to the smoothing patterns observed in vanilla GNN, other graph learning models such as APPNP [17] can also converge to a steady state. The converged state of APPNP is expressed as:

$$Z^{\infty} = \alpha(I_n - (1-\alpha)A)^{-1}X,$$

where $\alpha$ is the teleport probability. To investigate whether any smoothing pattern can be utilized for detecting anomalous nodes, we modify the graph convolution operation in SSGNN with APPNP update rule $Z^t = (1-\alpha)AZ^{t-1} + \alpha X$, and replace $B^t$ with $Z^t - Z^{\infty}$. The results of this modified model, denoted as SmoothGNN-A, are shown in Table 5. Due to the high computational complexity of the inversion of a matrix, we only report 8 datasets for SmoothGNN-A. We observe that by employing alternative smoothing patterns, the framework can still effectively detect anomalous nodes, thus validating that smoothing patterns serve as accurate identifiers for NAD. However, based on the comparison between SmoothGNN and SmoothGNN-A in Table 5, we find SmoothGNN can achieve relatively better performance in most datasets. These results demonstrate information from spectral space is also important in NAD.

## 6 Conclusion

In this paper, we introduce the individual and neighborhood smoothing patterns into the NAD task. We identify differences in the smoothing patterns between anomalous and normal nodes and further demonstrate the observation through comprehensive experiments and theoretical analysis. The combination of four components in SmoothGNN enables the model to capture information from both the spectral space and smoothing patterns, providing comprehensive perspectives for NAD tasks. Extensive experiments demonstrate that SmoothGNN consistently outperforms state-of-the-art competitors by a significant margin in terms of performance and running time, thus highlighting the effectiveness and efficiency of leveraging smoothing patterns in the NAD area.

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

# A Appendix

## A.1 Proofs

**Table 6: Frequently used notations**

| Notations | Descriptions |
|---|---|
| $G$ | The input graph. |
| $n, m$ | The number of nodes and edges. |
| $\delta, T$ | Threshold of Preprocess and layers of model. |
| $A, X, L$ | Adjacent, feature, and Laplacian matrix. |
| $P, B$ | Propagation and augmented propagation matrix. |
| $V_a, V_n$ | The set of anomalous and normal nodes. |
| $I(x), N(x)$ | Individual and neighborhood smoothing patterns. |

Table 6 lists the notations that are frequently used in this paper.

## A.2 Proofs

**Proof of Theorem 1.** The following lemma from previous work [3] is used for the proof.

LEMMA 1. *Let $P = \frac{I_n}{2} + \frac{\tilde{A}}{2}$ denote the propagation matrix given the adjacency matrix $\tilde{A}$, we have:*

$$P_{i,j}^\infty = \frac{\sqrt{d_i+1}\sqrt{d_j+1}}{2m+n}.$$

First, we need to derive $P^t - P^\infty = (P - P^\infty)^t$. For $t \geq 1$, we have:

$$(P - P^\infty)^t = \sum_{k=0}^{t} \binom{t}{k}(-1)^k P^{t-k} P^\infty$$

$$= P^t + \sum_{k=1}^{t} \binom{t}{k}(-1)^k P^\infty$$

$$= P^t + P^\infty((1-1)^n - 1) = P^t - P^\infty.$$

Then, we can derive that

$$B_{i,j} = P_{i,j} - P_{i,j}^\infty$$

$$= \frac{(2m+n)(\mathbb{I}[i=j]\sqrt{d_i+1} + 2a_{ij}) - 2(d_i+1)\sqrt{d_j+1}}{2\sqrt{d_i+1}(2m+n)}.$$

Then Theorem 1 can be proved. □

**Proof of Theorem 2.** Based on the proof in Section "The stable distribution" in the book [4], for the column vector $b^t$ in augmented matrix $B^t$ we have:

$$b^t = D^{\frac{1}{2}} \sum_{i=2}^{n} \omega_i^t c_i \psi_i,$$

where $D$ is the degree matrix of graph $G$, $\omega_i$ is the $i$-th eigenvalue of $P$, $\psi_i$ is the $i$-th eigenvector of $\tilde{A}$, and $c_i$ is the coefficient related to $\psi_i$. Then, let $\lambda_i^L$ be the $i$-th eigenvalue of $L$ and $\lambda_i^A$ be the $i$-th eigenvalue of $\tilde{A}$, we have:

$$\lambda_i^L = 1 - \lambda_i^A = 1 - (2\omega_i - 1) = 2 - 2\omega_i.$$

Then we replace $\omega_i$ in $D^{\frac{1}{2}} \sum_{i=2}^{n} \omega_i^t c_i \psi_i$ with $\lambda_i^L$, we further have $D^{\frac{1}{2}} \sum_{i=2}^{n} (\frac{\lambda_i^L}{2} - 1)^t c_i \psi_i$. By applying Tayler's expansion to it, we have:

$$\sum_{i=2}^{n} (\frac{\lambda_i^L}{2} - 1)^t = \sum_{i=2}^{n} \sum_{k=0}^{t} \frac{t!}{(k-1)!(t-(k-1))!} (\frac{\lambda_i^L}{2})^{k-1} (-1)^{t-(k-1)}$$

$$= \sum_{k=0}^{t} \sum_{i=2}^{n} \frac{t!}{(k-1)!(t-(k-1))!} (\frac{\lambda_i^L}{2})^{k-1} (-1)^{t-(k-1)}$$

$$= \sum_{k=0}^{t} \frac{(\frac{1}{2})^{k-1} (-1)^{t-(k-1)} t!}{(k-1)!(t-(k-1))!} \sum_{i=2}^{n} (\lambda_i^L)^{k-1}$$

$$= \sum_{k=0}^{t} \theta_k \Lambda^k \mathbf{1} = \sum_{k=0}^{t} \theta_k U^T U \Lambda^k U^T U \mathbf{1} = \sum_{k=0}^{t} \tilde{\theta}_k L^k \boldsymbol{u}.$$

Finally, we can get

$$\boldsymbol{b}^t = D^{\frac{1}{2}} \sum_{i=2}^{n} (\frac{\lambda_i^L}{2} - 1)^t c_i \psi_i$$

$$= \sum_{k=0}^{t} \tilde{\theta}_k L^k \boldsymbol{u} \boldsymbol{v}.$$

This finishes the proof of Theorem 2. □

**Proof of Theorem 3.** For simplicity, let $\boldsymbol{x}$ denote a normalized graph signal in the graph, we have:

$$N(\boldsymbol{x}) = \sum_{i,j=1}^{n} a_{i,j} \| \frac{x_i}{\sqrt{d_i + 1}} - \frac{x_j}{\sqrt{d_j + 1}} \|_2^2$$

$$= \boldsymbol{x} L \boldsymbol{x}.$$

Following the theorem in previous work [6], we have:

$$\sum_{j=1}^{n} \lambda_j \hat{x}_j^2 = \boldsymbol{x}^T L \boldsymbol{x}.$$

Then Theorem 3 can be proved. □

**Proof of Theorem 4.** The following corollary from previous work [23] is used for the proof.

COROLLARY 1. *Let $\lambda_1 \leq \cdots \leq \lambda_n$ be the eigenvalues of $P$, sorted in ascending order. Suppose the multiplicity of the largest eigenvalue $\lambda_n$ is $m(\leq n)$, i.e., $\lambda_{n-m} < \lambda_{n-m+1} = \cdots = \lambda_n$. And the second largest eigenvalue can be defined as*

$$\lambda := \max_{s=1}^{n-m} |\lambda_s| < |\lambda_n|.$$

*Let $U$ be the eigenspace associated with $\lambda_n$, then we can assume that $U$ has an orthonormal basis that consists of non-negative vectors, and then we have:*

$$d_S(H^t) \leq \tau_t \lambda d_S(H^{t-1}),$$

*where $\tau_t \lambda < 1$ implying the output of the $t$-th layer of GNN on $G$ exponentially approaches $S$.*

Based on the above corollary, we have:

$$d_S(H^t) \leq \tau_t \lambda d_S(H^{t-1})$$

$$\leq (\prod_{i=1}^{t} \tau_i) \lambda^t d_S(X)$$

$$\leq \tau^t \lambda^t d_S(X).$$

When the GNN reaches $\epsilon$-smoothing, we have:

$$d_S(H^t) \leq \tau^t \lambda^t d_S(X) \leq \epsilon \rightarrow t \log \tau \lambda < \log \frac{\epsilon}{d_S(X)}.$$

Since $0 \leq s\lambda < 1$, then we have $\log s\lambda < 0$, we have:

$$t > \frac{\log \frac{\epsilon}{d_S(X)}}{\log \tau \lambda}.$$

This finishes the proof of Theorem 4. □

## A.3 Observations

*Normal Nodes* ———□———     *Anomalous Nodes* ———△———

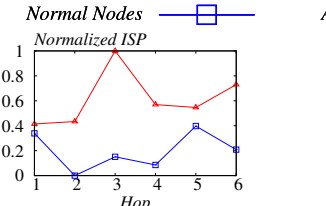
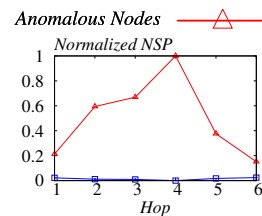

(a) Individual Smoothing Patterns   (b) Neighborhood Smoothing Patterns

**Figure 4: Smoothing Patterns of Reddit.**

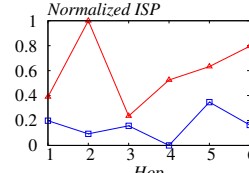
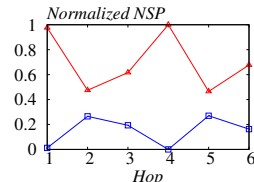

(a) Individual Smoothing Patterns   (b) Neighborhood Smoothing Patterns

**Figure 5: Smoothing Patterns of Tolokers.**

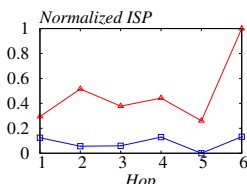
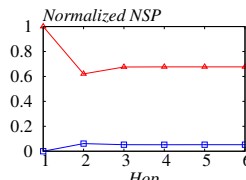

(a) Individual Smoothing Patterns   (b) Neighborhood Smoothing Patterns

**Figure 6: Smoothing Patterns of YelpChi.**

Observations from the additional datasets presented in Figures 4, 5, 6, 7, 8, 9, and 10 further reinforce our findings. These figures clearly show that the smoothing patterns of anomalous and normal nodes exhibit distinct trends and scales, where the ISP and NSP of anomalous nodes surpass those of normal nodes in most

**Normal Nodes** —□—    **Anomalous Nodes** —△—

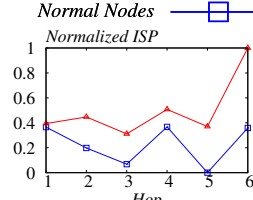
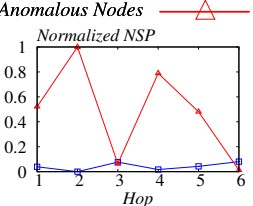

(a) Individual Smoothing Patterns    (b) Neighborhood Smoothing Patterns

**Figure 7: Smoothing Patterns of Questions.**

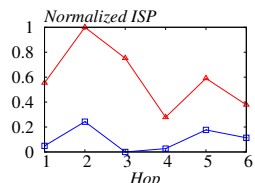
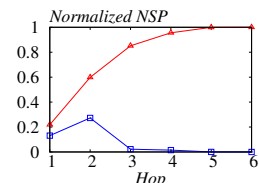

(a) Individual Smoothing Patterns    (b) Neighborhood Smoothing Patterns

**Figure 8: Smoothing Patterns of Elliptic.**

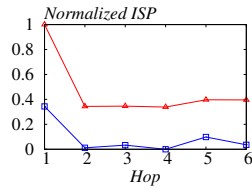
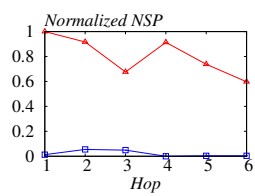

(a) Individual Smoothing Patterns    (b) Neighborhood Smoothing Patterns

**Figure 9: Smoothing Patterns of DGraph-Fin.**

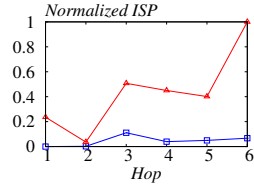
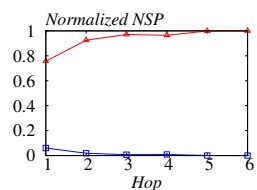

(a) Individual Smoothing Patterns    (b) Neighborhood Smoothing Patterns

**Figure 10: Smoothing Patterns of T-Social.**

cases. Our theoretical analysis and experimental results indicate that SmoothGNN is capable of detecting even subtle differences in these smoothing patterns. This sensitivity to nuanced smoothness characteristics is the key strength of the proposed approach.

## A.4 Algorithm

The detailed preprocess procedure is shown in Algorithm 1. Specifically, to facilitate the preprocess of large-scale graphs, we first apply an approximation technique to calculate the converged status of the propagation matrix, where $\delta$ represents the threshold as shown in Lines 1-6. Then, in Lines 7-10, we calculate the first $(T + 1)$ hops of the augmented propagation matrix to further reduce the running time cost during the training process.

Besides, the detailed training process is shown in Algorithm 2. We use the trained model of the final epoch to conduct inference.

The loss function is calculated using $H^{SCSLC}$ and $H^{SCGNN}$, and the anomaly score is calculated for each node using $f_{smooth}(\cdot)$ as shown in Section 4.5.

To be specific, in Lines 1-2, we calculate $(T + 1)$ transformations of node representations using a set of $(T + 1)$ MLPs. In Line 3, the SC is encoded into $\boldsymbol{\alpha}$ as shown in Section 4.4. After that, in Lines 4-10, we use the SLC and SSGNN components to calculate $\boldsymbol{h}_i^{SLC}$ and $\boldsymbol{h}_i^{GNN}$ for each node $i$ and apply the $\boldsymbol{\alpha}$ to serve as attention coefficients to rescale the learned representations. Notice that, to gain a superior reduction in terms of running time during the training process, we utilize the most simple architecture as the backbone. However, as shown in Section 4.1, we invent the framework from a distinct perspective, the smoothing pattern view, from all the previous works, which demonstrates the obvious differences between our work and previous ones. In Section 5, we conduct comprehensive experiments to prove the effectiveness and efficiency of our SmoothGNN.

---

**Algorithm 1:** Preprocess

**Input:** $\tilde{A}, T, m, n, \delta$
**Output:** $[P^0, \cdots, P^T], [B^0, \cdots, B^T]$
1  $\boldsymbol{deg} \leftarrow \text{Degree}(\tilde{A})$;
2  $\boldsymbol{deg} \leftarrow \frac{\boldsymbol{deg}}{\sqrt{2m+n}}$;
3  **for** $i = 1$ *to* $n$ **do**
4      **if** $\boldsymbol{deg}_i \leq \delta$ **then**
5         $\boldsymbol{deg}_i \leftarrow 0$;
6  $P^\infty \leftarrow \boldsymbol{deg} \cdot \boldsymbol{deg}^T$;
7  $P \leftarrow \frac{I_n}{2} + \frac{\tilde{A}}{2}$;
8  $L \leftarrow I_n - \tilde{D}^{-\frac{1}{2}} \tilde{A} \tilde{D}^{-\frac{1}{2}}$;
9  **for** $t = 0$ *to* $T$ **do**
10     $B^t \leftarrow P^t - P^\infty$;
11 Return $[P^0, \cdots, P^T], [B^0, \cdots, B^T]$;

---

**Algorithm 2:** SmoothGNN

**Input:** $X, T, n, [P^0, \cdots, P^T], [B^0, \cdots, B^T]$
**Output:** $H^{SCSLC}, H^{SCGNN}$
1  **for** $t = 0$ *to* $T$ **do**
2      $\tilde{X}_t \leftarrow \sigma(\text{MLP}(X))$;
3  $\boldsymbol{\alpha} \leftarrow \sigma(\text{MLP}(\text{CAT}(SC(P^0\tilde{X}_0), \cdots, SC(P^T\tilde{X}_T))))$;
4  **for** $i = 0$ *to* $n$ **do**
5      $\boldsymbol{h}_i^{SLC} \leftarrow \text{MLP}(\text{CAT}((B^0\tilde{X}_0)_i, \cdots, (B^T\tilde{X}_T)_i))$;
6      $\boldsymbol{h}_i^{GNN} \leftarrow \text{MLP}(\text{CAT}((g(\tilde{X}_0)_0)_i, \cdots, (g(\tilde{X}_T)_T)_i))$;
7      $\boldsymbol{h}_i^{SCSLC} \leftarrow \boldsymbol{h}_i^{SLC} * \boldsymbol{\alpha}$;
8      $\boldsymbol{h}_i^{SCGNN} \leftarrow \boldsymbol{h}_i^{GNN} * \boldsymbol{\alpha}$;
9  $H^{SCSLC} \leftarrow [\boldsymbol{h}_1^{SCSLC}, \cdots, \boldsymbol{h}_n^{SCSLC}]$;
10 $H^{SCGNN} \leftarrow [\boldsymbol{h}_1^{SCGNN}, \cdots, \boldsymbol{h}_n^{SCGNN}]$;
11 Return $H^{SCSLC}, H^{SCGNN}$;

**Table 7: Parameters for SmoothGNN according to different categories.**

| Categories | Learning Rate | Hop | Weight Initialization | Delta | Hidden Dimensions |
|------------|---------------|-----|-----------------------|-------|-------------------|
| Small | 1e-4 | 4 | 0.05 | 0 | 64 |
| Medium | 5e-4 | 5 | 0.01 | 4e-3 | 64 |
| Large | 5e-4 | 6 | 0.05 | 4e-3 | 64 |

**Table 8: $\frac{S_a - S_n}{S_n}$ for each 10 epoch, where $S_n$ and $S_a$ are the SMeasures of normal and anomalous nodes separately.**

| Datasets | 0 | 10 | 20 | 30 | 40 | 50 | 60 | 70 | 80 | 90 | 100 |
|----------|-----|-----|-----|-----|-----|-----|-----|-----|-----|-----|-----|
| Reddit | 0.0029 | 0.0410 | 0.0835 | 0.1000 | 0.1059 | 0.1105 | 0.1154 | 0.1193 | 0.1220 | 0.1237 | 0.1242 |
| Tolokers | 0.0722 | 0.2290 | 0.2883 | 0.2728 | 0.2704 | 0.2708 | 0.2665 | 0.2628 | 0.2631 | 0.2666 | 0.2719 |
| Amazon | -0.0461 | 0.0516 | 0.0923 | 0.0892 | 0.1160 | 0.1251 | 0.1260 | 0.1288 | 0.1325 | 0.1383 | 0.1460 |
| T-Finance | -0.0496 | 0.0004 | 0.0419 | 0.0858 | 0.1130 | 0.1253 | 0.1260 | 0.1234 | 0.1252 | 0.1254 | 0.1238 |
| YelpChi | 0.0353 | 0.0952 | 0.1049 | 0.1124 | 0.1140 | 0.1142 | 0.1122 | 0.1117 | 0.1103 | 0.1088 | 0.1073 |
| Questions | 0.0267 | 0.1211 | 0.1683 | 0.1822 | 0.2096 | 0.2351 | 0.2511 | 0.2650 | 0.2776 | 0.2915 | 0.3012 |
| Elliptic | 0.0036 | 0.1675 | 0.2300 | 0.2657 | 0.2858 | 0.2972 | 0.3038 | 0.3072 | 0.3084 | 0.3079 | 0.3059 |
| DGraph-Fin | -0.0727 | 0.1516 | 0.1519 | 0.1518 | 0.1511 | 0.1506 | 0.1500 | 0.1496 | 0.1491 | 0.1485 | 0.1474 |
| T-Social | -0.0026 | 0.0812 | 0.4438 | 0.4993 | 0.5765 | 0.6491 | 0.6986 | 0.7134 | 0.6800 | 0.6131 | 0.5334 |

## A.5 Experimental Settings

The parameters are set based on the number of nodes in different graphs as shown in Table 7. As we can see, the hidden dimensions remain stable for all three categories. However, the learning rate and number of propagation hops increase as the number of nodes grows. Besides, we employ approximation techniques to calculate the converged status of propagation, i.e., we only retain the values larger than the square of delta in the final matrix. For small graphs, we do not require approximation whereas for medium and large graphs, we set the delta to 4e-3. Furthermore, the weight initialization strategy varies across graph categories because the optimal starting point in the optimization process tends to differ depending on the graph characteristics. As for the experimental environment, we conduct all the experiments on CPUs to provide enough memory for previous works.

## A.6 SMeasure during training

To explicitly show why SMeasure can be used for unsupervised NAD tasks, we report $\frac{S_a - S_n}{S_n}$ for each 10 epoch, where $S_n$ and $S_a$ are SMeasures of normal and anomalous nodes separately. As observed

from Table 8, $\frac{S_a - S_n}{S_n}$ grows to a positive number, which means $S_a$ is larger and grows faster than $S_n$ as the training continues. These results tell us our framework can effectively capture the ISP and NSP through SMeasure and utilize the novel measure to effectively detect anomalous nodes.

## A.7 Limitations

Although SmoothGNN achieves outstanding performance in all 9 real-world datasets compared to previous works, it still has some aspects to be improved. First, in this paper, we only discuss the smoothing patterns of GNN and APPNP, while there are other kinds of models in the field of graph learning. The performance of different types of smoothing patterns can vary due to their ability to capture additional information, such as spectral space. Second, Compared to semi-supervised and supervised NAD tasks, the performance of SmoothGNN is still unsatisfactory. Hence, it is also interesting to employ smoothing patterns in semi-supervised and supervised NAD tasks. Third, we only present the effectiveness of smoothing patterns in the NAD area, but applying smoothing patterns to other related fields can be meaningful as well.

