# OpenReview forum: "SmoothGNN: Smoothing-aware GNN for Unsupervised Node Anomaly Detection"
_ACM.org/TheWebConf/2025/Conference — WWW 2025 Oral_

### Official Review · Reviewer_MD9z · 2024-11-30

**Novelty:** 6
**Technical Quality:** 6

**Review:**

The paper introduces SmoothGNN, a novel method for unsupervised node anomaly detection in graphs using smoothing patterns and spectral graph theory. It proposes Individual Smoothing Patterns (ISP) and Neighborhood Smoothing Patterns (NSP) to detect anomalies with theoretical solid insights and experimental validation across multiple datasets. The results show that SmoothGNN outperforms state-of-the-art methods in accuracy and computational efficiency, particularly on large-scale datasets, with up to 75x faster speed than competitors.
However, there are a few areas that require improvement:
The paper could benefit from more precise explanations of the mathematical concepts, particularly the formulations of ISP and NSP, as these might require more work for non-experts to grasp fully.
While the experimental results are thorough, the authors should address potential real-world challenges, such as memory limitations when dealing with large graphs and possible overfitting issues.
The paper compares SmoothGNN primarily with GNN-based methods, but a comparison with non-GNN-based anomaly detection techniques would offer a more comprehensive and balanced evaluation.
The authors should discuss strategies for preventing overfitting, especially given graph neural networks' complexity and tendency to overfit on small or imbalanced datasets.
In conclusion, SmoothGNN significantly contributes to graph anomaly detection with strong experimental results. However, additional clarification of the theoretical aspects and a broader discussion of its limitations and comparisons with non-GNN methods would significantly enhance the paper's overall quality and impact.

**Questions:**

How do Individual Smoothing Patterns (ISP) and Neighborhood Smoothing Patterns (NSP) behave in graphs with varying structures (e.g., sparse vs. dense)? How do the graph’s spectral properties influence these patterns and the robustness of SmoothGNN?

How do you determine the optimal number of propagation hops for different datasets? Is this choice based on theory or empirical results, and how does SmoothGNN address the risks of over-smoothing or under-smoothing with too many or too few hops?

**Reviewer Confidence:**

4: The reviewer is certain that the evaluation is correct and very familiar with the relevant literature

**Scope:**

4: The work is relevant to the Web and to the track, and is of broad interest to the community

---

### Official Review · Reviewer_W9sy · 2024-12-02

**Novelty:** 5
**Technical Quality:** 6

**Review:**

### Summary

This paper presents a theoretical analysis and a novel model for node anomaly detection (NAD). The proposed model, SmoothGNN, leverages theoretical properties referred to as ISP and NSP and demonstrates consistent performance improvements across various datasets.

### Strengths

S1. The paper introduces a novel model based on theoretical characteristics (ISP and NSP) that have been relatively underexplored in the NAD domain.

S2. Extensive experiments are conducted, showcasing performance improvements over numerous datasets and baselines.

### Weaknesses

W1. The claim that NSP serves as the coefficient for SmoothGNN layers lacks sufficient justification. The rationale for assigning NSP as a layer-wise coefficient should be clarified. Furthermore, it is unclear whether NSP could be applied only at the final layer representation instead of across layers. Experimental results exploring this alternative would be valuable.

W2. The utility of Theorem 4 in guiding the selection of the number of layers is unclear. It would be helpful to analyze whether the experimental outcomes align with the theoretical results derived in Theorem 4.

W3. A detailed analysis of time and space complexity is missing. The proposed model does not appear to be more efficient than existing GNN-based or linear models. Moreover, several cells in Tables 2 and 3 lack results, raising concerns about the comprehensiveness of the experimental evaluation.

**Questions:**

Please refer to the weaknesses above.

**Reviewer Confidence:**

3: The reviewer is confident but not certain that the evaluation is correct

**Scope:**

4: The work is relevant to the Web and to the track, and is of broad interest to the community

---

### Official Review · Reviewer_CgwL · 2024-12-03

**Novelty:** 4
**Technical Quality:** 3

**Review:**

The paper presents a novel framework for detecting anomalous nodes in graphs. The authors argue that the smoothing issue in graph learning, which typically hinders the distinguishability of node representations, can be leveraged to identify anomalies. They introduce the concepts of Individual Smoothing Patterns (ISP) and Neighborhood Smoothing Patterns (NSP) and demonstrate their effectiveness in unsupervised node anomaly detection (NAD) tasks.

Strengths:
1. The paper introduces a new perspective on using smoothing patterns for NAD, which views smoothing as a negative aspect of graph learning.
2. The proposed SmoothGNN framework combines ISP and NSP with spectral graph neural networks, offering an approach to NAD.
3. The authors provide experiments on 9 real-world datasets, demonstrating the effectiveness of SmoothGNN over state-of-the-art methods.

Weaknesses:
1. The paper claims a significant speed-up compared with other methods but does not provide a detailed analysis of the computational or time complexity. I am curious about how the proposed method achieves a relatively low time complexity given the 4 components involved. Besides, under what settings was Table 3 generated, it seems that this paper does not reference Table 3.
2. It is hoped that the code will be made open-source for careful review.

**Questions:**

1. Are there any particular types of datasets that are more "conducive" to SmoothGNN than others?
2. If the effectiveness of ISP and NSP relies on certain assumptions about the graph structure and node attributes? Cant it hold for all types of graphs.
3. Although the paper mentions that there are other kinds of models in the field of graph learning, it only briefly notes this issue without analyzing whether the smooth pattern can be applied to other models?

**Reviewer Confidence:**

3: The reviewer is confident but not certain that the evaluation is correct

**Scope:**

4: The work is relevant to the Web and to the track, and is of broad interest to the community

---

### Official Review · Reviewer_WH6K · 2024-12-03

**Novelty:** 5
**Technical Quality:** 4

**Review:**

Based on the observations that anomalous node representations are more difficult to smooth compared to normal ones, the paper introduces Individual Smoothing Patterns (ISP) and Neighborhood Smoothing Patterns (NSP) for node anomaly detection on graphs. This is interesting and is well supported by theoretical analysis.

**Questions:**

My concerns include:
- Regarding your loss function 𝐿 = 𝐿𝑐𝑜𝑛 + 𝐿𝑠𝑚𝑜𝑜𝑡ℎ, you assign weights of 1 to them. Have you considered incorporating a tradeoff hyperparameter, and what impacts would it have on these two components?
- In Table 2, where most AUC results fall below 80%, the detection performance appears to be akin to random guessing. Do these anomaly detection tasks present considerable challenges for learning?
- Section A.7 delves into potential limitations. It notes, "Compared to semi-supervised and supervised NAD tasks, the performance of SmoothGNN remains unsatisfactory." Could you provide more results of the semi-supervised and supervised tasks? Is the smoothing issue also observed in the semi-supervised and supervised cases?

**Reviewer Confidence:**

3: The reviewer is confident but not certain that the evaluation is correct

**Scope:**

4: The work is relevant to the Web and to the track, and is of broad interest to the community

---

### Official Review · Reviewer_QiMS · 2024-12-03

**Novelty:** 6
**Technical Quality:** 6

**Review:**

Strengths
1. The paper is well written with clear motivations.
2. The proposed method is reasonably designed to tackle the problem of graph anomaly detection. Detailed mathematical expressions are included.
3. The experimental results are promising with up-to-date baseline methods.

Weakness
1. In Table 2, it seems that there are several cases where the proposed method does not perform the best, such as on the Elliptic dataset, REC outperforms. More in-depth analysis is recommended, such as the property of the dataset, and why the baseline outperforms.
2. The theoretical computational complexity is not provided in the paper.

**Questions:**

1. What if the input graph is heterogeneous, does the proposed anomaly detection method still stand?
2. How to explain the time efficiency of the proposed method from the perspective of computational complexity?

**Reviewer Confidence:**

3: The reviewer is confident but not certain that the evaluation is correct

**Scope:**

4: The work is relevant to the Web and to the track, and is of broad interest to the community

---

### Official Review · Reviewer_7hVn · 2024-12-06

**Novelty:** 5
**Technical Quality:** 5

**Review:**

To be honest, I am not very familiar with this field.

I find the paper well-written, with rich content and comprehensive experiments.

**Questions:**

Why are there some missing values in the experimental result table?

**Reviewer Confidence:**

2: The reviewer is willing to defend the evaluation, but it is likely that the reviewer did not understand parts of the paper

**Scope:**

4: The work is relevant to the Web and to the track, and is of broad interest to the community